# Influenza Antigens NP and M2 Confer Cross Protection to BALB/c Mice against Lethal Challenge with H1N1, Pandemic H1N1 or H5N1 Influenza A Viruses

**DOI:** 10.3390/v13091708

**Published:** 2021-08-27

**Authors:** Nutan Mytle, Sonja Leyrer, Jon R. Inglefield, Andrea M. Harris, Thomas E. Hickey, Jacob Minang, Hang Lu, Zhidong Ma, Hanné Andersen, Nathan D. Grubaugh, Tina Guina, Mario H. Skiadopoulos, Michael J. Lacy

**Affiliations:** 1Emergent BioSolutions, 300 Professional Drive, Gaithersburg, MD 20879, USA; nutan.mytle@hhs.gov (N.M.); sonja.leyrer@gmx.net (S.L.); jon.inglefield@nih.gov (J.R.I.); andreamharris-22@outlook.com (A.M.H.); thomas.hickey@nih.gov (T.E.H.); jminang@ohc-inc.com (J.M.); hanglu6@yahoo.com (H.L.); taiyang@yahoo.com (Z.M.); grubaughlab@gmail.com (N.D.G.); guina.tina@gmail.com (T.G.); mskiadopoulos@hotmail.com (M.H.S.); 2Biomedical Advanced Research and Development Agency, U.S. Department of Health and Human Services, Washington, DC 20201, USA; 3Roche Diagnostics GmbH, Nonnenwald 2, 82377 Penzberg, Germany; 4Frederick National Laboratory for Cancer Research, Frederick, MD 21702, USA; 5National Cancer Institute, National Institutes of Health, Frederick, MD 20814, USA; 6Optimal Health Care, 11377 Robinwood Dr, Hagerstown, MD 21742, USA; 7BIOQUAL, Inc., 12301 Parklawn Dr, Rockville, MD 20852, USA; handersen@bioqual.com; 8Yale School of Public Health, Yale University, 60 College Street, New Haven, CT 06510, USA; 9AstraZeneca, Gaithersburg, MD 20878, USA; 10U.S. Department of Health and Human Services, National Institutes of Health, Bethesda, MD 20892, USA

**Keywords:** modified Vaccinia virus Ankara (MVA), influenza A, nucleoprotein, matrix protein 1, matrix protein 2, hemagglutinin, conserved antigens, vaccine

## Abstract

Influenza hemagglutinin (HA) is considered a major protective antigen of seasonal influenza vaccine but antigenic drift of HA necessitates annual immunizations using new circulating HA versions. Low variation found within conserved non-HA influenza virus (INFV) antigens may maintain protection with less frequent immunizations. Conserved antigens of influenza A virus (INFV A) that can generate cross protection against multiple INFV strains were evaluated in BALB/c mice using modified Vaccinia virus Ankara (MVA)-vectored vaccines that expressed INFV A antigens hemagglutinin (HA), matrix protein 1 (M1), nucleoprotein (NP), matrix protein 2 (M2), repeats of the external portion of M2 (M2e) or as tandem repeats (METR), and M2e with transmembrane region and cytoplasmic loop (M2eTML). Protection by combinations of non-HA antigens was equivalent to that of subtype-matched HA. Combinations of NP and forms of M2e generated serum antibody responses and protected mice against lethal INFV A challenge using PR8, pandemic H1N1 A/Mexico/4108/2009 (pH1N1) or H5N1 A/Vietnam/1203/2004 (H5N1) viruses, as demonstrated by reduced lung viral burden and protection against weight loss. The highest levels of protection were obtained with NP and M2e antigens delivered as MVA inserts, resulting in broadly protective immunity in mice and enhancement of previous natural immunity to INFV A.

## 1. Introduction

Despite the high rates of spontaneous mutation among RNA viruses, a rate close to the maximum value compatible with viability [1], control of disease from INFV, an RNA virus, is attempted by vaccination. Currently licensed influenza vaccines induce subtype-specific virus-neutralizing antibodies with specificity for the densely expressed surface antigens neuraminidase (NA) and hemagglutinin (HA), of which eighteen HA subtypes and eleven NA subtypes have been identified.

Seasonal influenza vaccines that are based upon matching to current HA types, have limitations, examples of which are: (a) a low pooled efficacy of 59% in adults aged 18–65 years [2]; (b) only 24% cross reactivity based upon seasonal influenza vaccine (2009–2010) for seroprotective HA-based titer to pandemic 2009 swine H1N1 influenza virus [3]; and (c) escape mutations, for example 42.9% of individuals who succumbed to pandemic influenza in 2009 had been infected with INFV that had one of two identified HA mutations [4] that may have assisted immune escape. Some INFV proteins have lower mutation rates compared to HA. The INFV matrix protein gene [5] has a lower rate of nucleotide substitution compared to HA [6]. Similarly, the amino acid substitution rate per amino acid per year is 0.00074 for INFV NP [7] compared to 0.006 for human INFV A HA.

More conserved protein antigens of INFV A or conserved regions of dominant proteins [8,9,10] maintain T-cell responses [11,12] as well as antibody responses that may provide broad immunity [13]. Vector delivery systems also may improve vaccine efficacy, such as modified Vaccinia virus Ankara (MVA), which is a replication-deficient poxvirus strain that generates T-cell as well as antibody responses [14,15] recommended for influenza protection [16]. For example, MVA that expressed NP with or without M1 [17,18,19], protected mice against INFV infection and was safe in human phase I clinical trials [20,21,22].

In the present study, MVA-vectored vaccines expressed INFV antigens HA, NP, M1, M2, M2eTML or METR as single antigens that were delivered separately or in combination (blend). One MVA candidate vaccine was constructed to simultaneously express three antigens (M1, METRC and NP). Efficacy was tested against homologous and heterologous INFV A virus. Heterology was based upon (a) H3N2 origin of NP within the vaccines but challenge using PR8 H1N1, pH1N1 or H5N1, and (b) priming using PR8 H1N1 live virus followed by heterologous vaccine boosting to achieve protection against subsequent heterologous H5 challenge. The resulting data verified that conserved, non-HA influenza A antigens can be appropriate for a universal influenza vaccine.

## 2. Materials and Methods

### 2.1. Experimental Design

Vaccines were administered as two intramuscular (IM) injections of 75 µL into each thigh (total 150 µL) on study days 0 and 18–22 to deliver an MVA TCID50 value of 5 to 8 × 10^7^ per mouse. Sera were obtained on day 0, three weeks after the first immunization, and three weeks after the second immunization. Selected sera were collected 14 days after challenge.

Survival, weight loss, reduction in the viral burden, antibody responses and cellular responses were assessed.

### 2.2. Animals

Animal studies were conducted at BIOQUAL, Inc., Rockville, MD, USA, in compliance with the Animal Welfare Act and following principles outlined in the National Research Council Guide for the Care and Use of Laboratory Animals. All animal study protocols (No. 10-3478-63; 14-3478-33) were approved by BIOQUAL’s Institutional Animal Care and Use Committee (IACUC). All studies were conducted in an animal biosafety level-2 (ABSL-2) facility in accordance with USDA guidelines for use of MVA in animals, and the challenge with H5N1 was performed in a Select Agent-approved animal biosafety level-3 (ABSL-3) facility. All experiments were conducted using female BALB/c mice purchased from Harlan (Frederick, MD, USA) acclimated in ABSL-2 containment for 1 week prior to study initiation. Following IACUC guidance, mice were euthanized using procedures in accordance with AVMA Guidelines for the Euthanasia of Animals: 2013 Edition when reaching 20% weight loss after viral challenge.

### 2.3. Vaccine Preparations

INFV A antigens were inserted by homologous recombination into the MVA vector: hemagglutinin (HA) of H1N1 A/PR/8/34 (PR8HA) or H5N1 A/VN/1203/04 virus (H5HA); matrix protein 1 (M1) of H1N1 A/Brisbane/59/2007; nucleoprotein (NP) consensus sequence of 700 NP human INFV A from 2004 to 2010, with up to 100% amino acid homology with H3N2 viruses and 80–95% amino acid homology to H1N1 NP protein; matrix protein 2 (M2) of H1N1 influenza A/PuertoRico/8/1934, 97 amino acids; matrix protein 2 extracellular domain plus transmembrane domain plus internal loop (M2eTML) of H1N1 A/PuertoRico/8/1934, 58 amino acids; M2 external domain (ME) tandem repeats (TR) containing 6 different ME sequences (METR), each of which contained cysteine amino acids (METRC) (Figure 1) or cysteine replaced with serine amino acids (METRS). MVA without INFV antigen (MVAtor) or PBS was used as a control. MVA constructs contained a single antigen (HA, NP, M1, M2, M2eTML, METRC, or METRS) or three antigens (M1, METRC, and NP) in a single insertion site, each antigen sequence with a promoter. Sequence was confirmed by nucleic acid sequencing, expression was confirmed by immunofluorescence as described [23] and molecular weight was confirmed by immunoblot of cell lysates under reducing conditions. Flulaval^®^ (GSK, CA/07/2009(H1N1), A/Victoria/210/2009(H3H2), B/Brisbane/60/2008) was administered 2 × 75 µL in each thigh resulting in 4.4 µg HA per INFV strain.

### 2.4. Challenge

H1N1/A/Puerto Rico 8/1934 (PR8) virus was purchased from ATCC (Manassas, VA, USA) and expanded using 9- to 11-day-old specific pathogen-free embryonated chicken eggs (Charles River, North Franklin, CT, USA) producing a viral stock with infectious titer of 10^7.2^ times the 50% tissue culture infectious dose per mL (TCID_50_/mL). The 50% lethal dose (LD_50_) of each of PR8 and the other challenge viruses was confirmed prior to the initiation of efficacy studies (data not shown). Mice were challenged intranasally with 629 TCID_50_ (3 LD_50_ in 50 µL) of the PR8 challenge virus.

Pandemic swine influenza H1N1 A/Mexico/4108/2009 (pH1N1) virus stock was generated at BIOQUAL, Inc. (Rockville, MD, USA) in 10-day-old embryonated chicken eggs using a seed stock obtained from the U.S. Centers for Disease Control and Prevention (CDC, Atlanta, GA, USA) with an infectious titer of 10^6.25^ TCID_50_/mL. Mice were challenged intranasally with 50 µL of 10^5.62^ TCID_50_/mL (20 LD_50_) of the challenge pandemic virus.

H5N1 A/Vietnam/1203/2004 (H5N1) INFV A working stock was generated at BIOQUAL, Inc. (Rockville, MD, USA) in 10-day-old embryonated chicken eggs using a seed stock obtained from the CDC, to contain an infectious titer of 10^9.0^ TCID_50_/mL. Mice were challenged intranasally with 10^2.4^ TCID_50_ (10 LD_50_ in 50 µL) of the challenge virus.

Mice were challenged intranasally (50 µL total volume) 3 weeks after the second immunization. Clinical signs and weights were monitored daily for 14 days after virus challenge. In accordance with IACUC policy, loss of 20% or more of prechallenge body weight prompted humane euthanasia of the mice.

In one H5N1 challenge study, half of the mice were primed by IN delivery of sublethal PR8 at 200–315 TCID_50_ (<½ LD_50_) as an analog of natural mild INFV infection which was confirmed by 2% average weight loss on days 3 and 4 after challenge.

### 2.5. Recombinant Proteins for ELISA

Recombinant influenza NP of PR8 H1N1 was produced in SF9 insect cells using bacculovirus expression after PCR amplification of PR8 H1N1. M2 of PR8 H1N1 was expressed in *E coli* predominantly in tetramer form. M1 of A/Brisbane/59/2007 was expressed in *E coli*. Purity of these proteins was 100% assessed by densitometry of Coomassie stained reducing SDS PAGE.

### 2.6. Hemagglutination Inhibition (HAI) Assay

One part serum and three parts receptor-destroying enzyme (Accurate Chemicals, Westbury, NY, USA) were incubated overnight at 37 °C followed by heat inactivation for 30 min at 56 °C and serial dilution into PBS that contained 0.5% BSA. INFV A (25 µL) at a concentration of 8 hemagglutinating units per 50 µL and sera (25 µL) were combined in V-bottom 96-well plates and incubated for 30 min at room temperature (RT) followed by an equal volume (50 µL) of 1% turkey whole blood (Innovative Research, Novi, MI, USA) as a source of red blood cells. After incubation at RT for 60 min, the reciprocal of the serum dilution that blocked hemagglutination was obtained and reported after triplicate testing as the mode (most frequent value) per mouse or if no mode the intermediate value per mouse, and median values per group.

### 2.7. Tissue Culture Infective Dose 50% (TCID_50_) and Viral Burden

PR8 virus TCID_50_ infectivity value and lung tissue viral burden were determined as endpoint serial dilutions of virus or tissue homogenates at which 50% of the cells were infected using Madin-Darby canine kidney (MDCK) cells. Lungs obtained from animals 2, 3, or 4 days after INFV A exposure were homogenized using an Omni TH Homogenizer (Omni International, Kennasaw, GA, USA) and Omni tissue disposable generators in Lebovitz L-15 medium (Gibco Life Technologies, Grand Island, NY, USA) containing antibiotic-antimycotic mixture (Invitrogen, Grand Island, NY, USA). After clarification by centrifugation, supernatant was added (20 µL/well) to 4 replicate wells of a 96-well flat-bottomed tissue culture plate (Corning, Tewksbury, MA, USA) containing MDCK cells in logarithmic (log) phase growth, and serially diluted 10-fold for a total of 8 dilutions. The plates were incubated at 37 °C in 5% CO_2_ for 4 days and cells inspected microscopically. Cytopathic effect was noted and confirmed visually by adherence of turkey RBC. The results of viral load assessment were expressed as TCID_50_ per gram of tissue [24].

To quantify lung viral load, four mice per group were euthanized on day 3 following PR8 H1N1 virus challenge and on days 2 and 4 following pH1N1 or H5N1 virus challenge.

### 2.8. Enzyme-Linked Immunosorbent Assays (ELISA)

IgG antibody responses were quantified using ELISA plates (Nunc MaxiSorp, San Diego, CA, USA) coated with M1 or M2 protein (1 µg/mL) in PBS with 0.05% sodium azide (PBSA). After washing with PBS that contained 0.05% Tween-20 (PBST) and blocking using PBST with 0.5% BSA and 100 μg/mL thimerosal (PBSTBT), serum samples were serially diluted into PBSTBT, incubated for 4 h at RT, washed followed by horseradish peroxidase (HRP) conjugated goat anti-mouse IgG (Kirkegaard and Perry, Gaithersburg, MD, USA) for 1 h at RT and washed. Color was developed using tetramethyl benzidine (TMB) (Kirkegaard and Perry, Gaithersburg, MD, USA) for 10 min and STOP solution (Kirkegaard and Perry, Gaithersburg, MD, USA). Absorbance at 450 nm was obtained using a Spectra Max Plus (Molecular Devices, Sunnyvale, CA, USA) plate reader equipped with SOFTMAX Pro 5.4 software followed by log-log linear regression analysis. Standard curves utilized purified monoclonal antibody (mAb) anti-M1 B313M (Meridian Life Sciences, Memphis, TN, USA) or purified mAb anti-M2 14C2 (AbCam, Cambridge, MA, USA).

For anti-NP IgG ELISA, plates were coated using 0.5 µg/mL recombinant NP and were blocked using PBSTBT that contained also 2% fetal bovine serum. Standard curve purified mAb was B249M (Meridian Life Sciences, Memphis, TN, USA).

Similarly, anti-M2e peptide IgG serum levels were quantified using plates coated with 1 µg/mL of Neutravidin (Thermo-Fisher Scientific, Waltham, MA, USA) to immobilize peptide number 3 (ProImmune, Oxford, UK), 0.5 μg/mL, listed in Figure 1, and a standard curve of mAB 14C2.

### 2.9. Enzyme-Linked Immunospot (ELISpot) Assay

Splenic cellular responses were quantified using a mouse IFN-γ ELISpot procedure. In the first of two T-cell analyses, recombinant MVA (4 × 10^7^ TCID_50_ per mouse) that encoded NP was administered intramuscularly (IM) to BALB/c. Spleens were harvested 47 to 48 days after the second immunization and stimulated using NP MHC Class I-restricted peptide TYQRTRALV or M2 peptide, and cellular responses quantified using a mouse IFN-γ ELISpot procedure. H-2d MHC class I-restricted MVA-derived peptides (SPGAAGWDL of MVA protein F2, and VGPSNSPTF of MVA protein E3) confirmed administration of MVA. In the second study, a recombinant MVA vector that encoded NP, M1, and M2 in a single insertion site (trivalent construct) was administered intramuscularly (9 × 10^7^ TCID_50_ MVA per mouse) to 15 BALB/c mice. Spleens harvested two weeks after the second immunization were analyzed for MHC class I- and class II-restricted peptide-induced T-cell responses using mouse IFN-γ ELISpot.

ELIspot plates (BD BioSciences, San Jose, CA, USA) precoated with mouse IFNγ-specific antibody were used to test proteins and peptides (6 µg/mL, ProImmune, Oxford, UK). Mouse spleen cells were purified using Lympholyte M (Cedarlane Laboratories, Burlington, NC, USA) and plated at a density of 300,000 cells per well, 3 wells per mouse per recall antigen. Concanavalin A (Sigma, St. Louis, MO, USA) was positive control. Plates were incubated for 20 to 24 h, color developed using HRP-labeled paired antibody (BD Biosciences, San Jose, CA, USA) and TrueBlue substrate (Kirkegaard and Perry, Gaithersburg, MD, USA) and spots quantified using an ELISpot plate reader (Cellular Technology Limited, Shaker Heights, OH, USA).

### 2.10. Ethical Approval

All experiments using animals were conducted at BIOQUAL, Inc. under IACUC-approved animal use protocols: No. 10-3478-63; 14-3478-33. This study was conducted in accordance with animal welfare requirements and accreditations. Housing and handling of the animals were performed in accordance with the standards of the AAALAC International, the Animal Welfare Act as amended, and the Public Health Service Policy. Handling of samples and animals occurred in compliance with the CDC regulations on Biosafety in Microbiological and Biomedical Laboratories (BMBL).

## 3. Results

### 3.1. Survival and Body Weights Following INFV A Challenge

The INFV challenge strategy included initial testing of individual antigens followed by bivalent combinations and moving to the most likely trivalent vaccine. Multiple challenge INFV A were used to assess the progression of these antigen combinations.

Mice in negative control groups that received PBS or MVAtor succumbed to lethal PR8 H1N1 or H5N1 challenge but 3 of 16 negative control mice survived pH1N1 challenge.

Individual INFV antigens except M1 provided 50% to 100% protection from mortality against PR8 H1N1, pH1N1, or H5N1 challenge as summarized in Table 1. Mice that received MVA-M1 had low (38%) survival following pH1N1 challenge. The outcome improved (88% survival) for mice that received the combination vaccine M1 + NP. M1 was evaluated further for possible immune incompatibility due to interference in multivalent vaccines.

Vaccination with MVA-NP alone produced 100% survival in animals challenged with PR8 H1N1. Combinations of influenza proteins M2 and NP also protected 100% of mice, prompting testing of trivalent combinations of antigens expressed from a single genetic insert or as blended multiple individual vaccines.

To understand capability of boosting natural immunity, trivalent vaccines were administered after PR8 infection. All mice primed with PR8 viral infection were 100% protected from challenge with pandemic H5N1 regardless of the subsequent antigen combinations (Table 1). For example, recipients of the vaccine that expressed M1 + METRC + NP from a single insert (no natural priming) had 50% survival after H5N1 challenge, but 100% survival with PR8 priming.

Due to high levels of protection by these vaccines, survival alone did not provide a sensitive comparison. Weight loss was a more sensitive indicator of protection than survival.

In PR8 H1N1 challenge studies, separate antigens (M2, M2TML, METRC, METRS, or NP) resulted in weight loss >7% but NP antigen combined with any of M2, M2TML, METRC or METRS reduced weight loss to <7% as shown in Figure 2.

Further analysis of antigen combinations utilized pH1N1 challenge after administration of MVA-M1 alone or in combination (Figure 3). Despite substantial weight loss in recipients of single antigen MVA-M1 vaccine, M1 did not lessen the benefit of blended MVA M1 + MVA NP + MVA M2 or blended MVA M1 + MVA NP + MVA METRC vaccines.

Weight loss after H5N1 challenge clearly displayed the capability of MVA vaccines to boost immunity of previous influenza infection (Figure 4). For example, mice immunized with a trivalent “blend” (MVA M1 + MVA METRC + MVA NP) vaccine without PR8 priming suffered greater weight loss (15%) than matched controls (5%) that received PR8 priming. Curiously, mice that received the trivalent blend vaccine had lower weight loss than recipients of the trivalent single insert vaccine (MVA-M1 + METRC + NP). Ultimately, each trivalent vaccine resulted in low weight loss only slightly greater (≤5% for only 2 days) than homologous H5 HA MVA vaccine.

Therefore, weight loss analysis of disease severity showed the superiority of antigen combinations as well as absence of interference by M1.

### 3.2. Viral Burden in Tissues

Lung viral burden was reduced by MVA vaccines that expressed METRS, METRC, M2 or combination of NP + METRC, NP + METRS, and NP + M2 by approximately 10-fold compared to negative controls on day 3 following PR8 H1N1 challenge (data not shown) which was not significant. In a separate study, after H5N1 challenge, lung viral burden in recipients of the single insert M1 + NP + METRC vaccine or a blend of three MVA vaccines (M1 + METR + NP) were not significantly different (*p* = 0.85, 0.37, respectively) from MVAtor controls (Figure 5). In contrast, priming with PR8 virus as a pre-immunization analog of natural infection resulted in significantly lower mean lung viral burdens in MVAtor control groups compared to MVAtor recipients without priming (*p* = 0.0117) and in MVA single insert (M1 + NP + METRC) recipients compared to those without priming (*p* = 0.0161) (Figure 5). The low lung viral burden of recipients of MVA single insert vaccine (M1 + METRC + NP) that followed PR8 priming was below the level of detection (1.7 × 104 log TCID_50_/g; geometric mean) on day 4 following H5N1 challenge, the same level as observed in mice vaccinated with H5HA (Figure 5).

### 3.3. Antibody Responses to HA, NP, M2, M2e and M1 Antigens

Antigens encoded in MVA were immunogenic, leading to antibody responses. HAI using PR8 virus confirmed that antibody responses to HA encoded in MVA vaccines was similar to HA immunogenicity of PR8 virus (Table 2). Twenty-one days after two immunizations, mice that had received MVA PR8 HA vaccine produced HAI titers similar to mice that had received two sublethal doses of PR8 virus. Repeated IM administration of MVA PR8 HA vaccine or sublethal PR8 inoculation generated similar antibody responses based upon HAI testing (Table 2).

MVA vaccines generated strong antibody responses in mice directed to non HA antigens NP, M2, and M2e. In contrast, weak antibody responses were developed to M1 despite previous confirmation by immunoblot of expression of M1 proteins within DF-1 cells by MVA that encoded M1 (data not shown). In two separate studies, IgG anti-M1 serum levels remained low, 1.2 ± 2.7 µg/mL to 1.8 ± 3.5 µg/mL (group mean), even after two immunizations. Increased IgG anti-M1 levels were observed, however, in mice subjected to a previous sublethal exposure to PR8. In those mice, PR8 priming followed 21 days later by one administration of MVA M1 vaccine resulted in IgG anti-M1 levels of 12.3 to 21.3 µg/mL potentially due to priming rather than boosting with MVA M1 vaccine. However, the IgG anti-M1 response was boosted to 22.4 to 34.1 µg/mL after the second MVA M1 immunization (data not shown graphically).

Strong anti-NP IgG responses were formed in response to MVA-NP vaccines in these studies. Groups of mice that received MVA-NP alone or in combination with other antigens achieved serum levels of anti-NP IgG that exceeded 1 mg/mL after the second immunization regardless of the number of components in the vaccine or the manner in which they were combined (i.e., a single MVA vector or a mixture of single-antigen constructs). For example, after two immunizations, a second antigen (M2, M2eTML, METRC, or METRS) in an NP-containing vaccine induced serum levels of anti-NP IgG that were 1.1- to 1.5-fold the serum levels induced by NP alone in three comparisons and 2.4-fold higher in a fourth comparison. In addition, anti-NP IgG levels illustrated the priming effect of influenza infection (PR8). Indeed, vaccine-induced anti-NP IgG serum levels increased to 3.5–3.6 mg/mL due to PR8 priming, a 3.4- to 5.5-fold increase compared to no PR8 priming. At the same timepoint (day 59), PR8-primed-MVAtor recipients had 2.1 mg/mL IgG anti-NP serum levels, a factor of 1.05 compared to the previous serum level of 2.0 mg/mL on day 43 (Figure 6).

M2 antigens were also immunogenic. Due to their design, administration of METRC vaccines or METRS vaccines resulted in exposure of mice to six different variants of M2e (Figure 1). Anti-M2 IgG serum responses to all METRC vaccines (83.9 ± 83.6 µg/mL) used in the PR8 H1N1 study were higher (*p* < 0.001) than all M2 or M2eTML vaccines (14.9 ± 17.0 µg/mL) after two immunizations. In comparison, in the pH1N1 study, anti-M2 IgG serum responses to a blended MVA M1 + MVA NP + MVA METRC vaccine (123.8 ± 125.2 µg/mL) were lower (*p* = 0.02) than MVA M1 + MVA NP + MVA M2 vaccine (229.3 ± 141.7 µg/mL). Interestingly, Flulaval did not generate an anti-M2 IgG response (0.8 ± 0.7 µg/mL) compared to all negative controls (1.3 ± 3.2 µg/mL).

Priming using low-grade natural PR8 infection resulted in very high anti-M2 IgG responses (Figure 7) that otherwise were low using MVA METRC vaccines. For example, MVA that expressed M1 + METRC + NP in a single insert generated low anti-M2 IgG responses with one (1.6 ± 1.3 µg/mL) or two immunizations (33.3 ± 39.3 µg/mL, respectively). Despite this setting of low immunogenicity, weak previous infection by natural virus PR8 provided superior priming of the immune system to subsequent MVA M1 + METRC + NP (single insert) generating 916 ± 1194.7 ug/mL serum IgG anti-M2 that was not increased (910.5 ± 952.9 µg/mL) by a second immunization of single insert MVA M1 + METRC + NP. PR8 infection (PR8 prime followed by MVAtor immunizations) resulted in IgG anti-M2 serum levels of 11.7 ± 15.3 µg/mL rising only to 21.2 ± 23.2 µg/mL. Natural infection by INFV A, therefore, established the potential for strong recall IgG anti-M2 responses that otherwise would remain low in response solely to MVA METRC trivalent vaccine.

Biotinylated synthetic peptide SLLTEVETPIRNEWGCRCNDSSD of M2e (peptide #3 of Figure 1) was used to assess anti-M2 antibody responses also. In two separate studies, serum levels of IgG anti-peptide were consistent but low (6.5 to 8.8 µg/mL mean concentrations, 6 groups of mice) after two IM administrations of M2 or M2eTML vaccines but higher (20.7 to 54.3 µg/mL) after two IM administrations of METRC and highest (62.5 to 135 µg/mL) after two IM administrations of METRS. The relative contribution of each version of M2e to the total anti-M2 ELISA response was not quantified.

### 3.4. MHC Class I and Class II-Restricted T Cell Responses

NP peptide was a superior ELISpot marker of cellular immunity in BALB/c mice (five per group) that were immunized regardless of MVA-NP combination (MVA-NP + M1 + M2). NP MHC class I-restricted peptide TYQRTRALV [25,26] generated 325 ± 126 IFN-γ-positive spots per 300,000 cells 14 days after the second immunization (Table 3) compared to 3 spots for unstimulated cells, 98 ± 56 spots for NP protein, and 53 ± 31 spots using an NP Class II peptide RLIQNSLTIERMVLS [27]. Mean anti-NP IgG serum levels were 1537 ± 727 µg/mL at the same timepoint, verifying an anti-NP immune response.

M2e 9mer peptide VETPIRNEW [28] and the entire M2e peptide (SLLTEVETPIRNEWGCRCNGSSD) were weak recall antigens, with 38 ± 18 spots and 26 ± 18 spots per 300,000 cells (Table 3). Despite weak cellular immunity, anti-M2 IgG serum levels at the same timepoint were 320 ± 306 µg/mL, indicating antibody immunity to M2 protein. Full-length M2 protein or Class I peptides of M2 internal domains (negative controls) did not recall positive ELISpot results (10.6, 6.7 and 2.7 spots). Data are summarized in Table 3.

## 4. Discussion

MVA vectors that expressed conserved, non-HA influenza antigens protected BALB/c mice against lethal challenge of homologous and heterologous INFV A group 1 influenza viruses, PR8 H1N1, pH1N1 or H5N1. Relative protective capability as weight loss was more comparative than survival, resulting first in selection of bivalent NP with any M2 variation as potent vaccines, then verification that heterologous INFV infection (PR8) prior to immunization with trivalent M1 + NP + METRC provided weight loss protection nearly equivalent to HA vaccine matched to H5 challenge INFV. In addition, MVA vaccines resulted in strong antibody and cellular responses to NP, moderate antibody and cellular responses to M2 but weak antibody responses to M1 protein. The most striking result was obtained using an MVA vaccine that encoded M1 + NP + METRC as a single genetic insert, which was responsible for reducing H5N1 INFV lung burden to detection limits if administered as a boost to INFV infection priming.

M1 has been reported previously to be immunogenic in mice [30,31]. In this study, IgG anti-M1 serum levels were low (group means < 2 µg/mL) despite two administrations of MVA M1. Priming with sublethal PR8 H1N1 followed 21 days later by 2 immunizations 21 days apart using MVA-M1 vaccines resulted in >10-fold increases in mean IgG anti-M1 serum levels. The characteristics of M1 protein expressed in situ by MVA or epitopes display upon recombinant M1 ELISA antigen expressed in *E coli* are not known.

M2 antigens as METR vaccines, containing six natural variations of M2e, generated strong humoral immunity and provided 100% protection against H5N1 viral challenge as well as non-detectable lung viral burden if preceded by natural PR8 (H1N1) infection. Protection by M2 has been reported for identical branched [32] or multiple M2e protein sequences [33] or different M2e sequences [34,35,36] as well as combinations such as a recombinant M2e and NP fusion protein, each of H3N2 origin, that protected BALB/c mice from PR8 H1N1 challenge [37]. M2e lacks strong T-cell epitopes in the BALB/c model, therefore protection is due to anti-M2 antibodies which may include a mechanism of antibody-dependent cellular cytotoxicity (ADCC). Preliminary data were obtained using mouse antisera (anti-NP + METRC) with IgG anti-M2 content of 465 or 600 µg/mL and also which had low IgG1:IgG2a anti-M2 ratios (6:1 or 19:1), which were capable of ADCC attack of PR8-infected MDCK target cells. In contrast, high IgG anti-M2 content mouse sera (770 µg/mL) but with high IgG1:IgG2a anti-M2 content (194:1) did not elicit ADCC of PR8-infected MDCK target cells using an ADCC kit (Promega, Madison, WI, USA) (data not shown) that triggered a luminescent reporter element as a marker of cell death. Surface expression by MDCK cells of INFV antigens after infection is limited to proteins that have transmembrane regions which includes full-length M2 protein.

NP antigen generated strong antibody responses and strong ELISpot results. NP was found previously by others to elicit protective immunity in mice [38,39] as high dose (90 µg) H3N2 NP or vectored, which induced comparable antibody and T-cell responses and was protective against PR8 virus [40,41,42]. NP T-cell immunity [43] can protect mice but NP-specific CD8 cytotoxic T lymphocytes (CTL) alone that utilized epitope TYQRTRALV [44] did not protect mice. In one study, anti-NP monoclonal or polyclonal antibodies in combination with CD8 T cells were implicated in protection [45]. More specifically, isotype-switched anti-NP antibodies in immune complexes prolonged antigen presentation to Fc receptor-bearing dendritic cells thereby extending a CD8 response [46]. Supporting these findings, ELISpot analysis in this study found strong detection of an H-2d-restricted Class I NP peptide, and weaker generation of immunity to a Class II peptide of NP in parallel with strong IgG anti-NP serum levels.

The term “conserved” can be defined as amino acid sequences that are found consistently within analogous proteins. Vaccines that expressed conserved proteins M1, M2e and NP were tested in this study. For example, alignment of known T-cell [47,48,49] and antibody NP epitopes [50,51] of BALB/c (H-2d) mice found only 2 substitutions, both of which were Arg to Lys conserved substitutions.

In contrast, only 9 amino-terminal amino acids of the 24 amino acids of the M2e domain are conserved [32]. For two influenza strains used in this study compared to a representative Vietnam H5N1 strain (goose, ABF01883.1), five substitutions occurred in positions 11 through 24 of M2e, which included amino acid positions within published antibody epitopes of M2e, including mAbs NM2 [52], 8C6 [53], 14C2 [54], and MM2 [52]. In addition, the polyclonal antibody target sequence LTEVETPIRNEWGCRCN of M2e [54] includes only seven unsubstituted positions. In fact, the M2 gene has evolved at a rate similar to HA in human and swine lineages, but with a slow rate in avian lineages [55]. To overcome these variations, the METR vaccines used in these studies utilized six different M2e for broad coverage.

In contrast to inbred mice, the necessary number of human epitopes is expected to be large. An epidemiological simulation calculated that 13 ± 6 epitope clusters of influenza dominate within the global world population with an average duration of only 4 ± 2 years [56], which supports a paradigm for influenza vaccine design that uses multiple antigens that possess multiple conserved epitopes.

## Figures and Tables

**Figure 1 viruses-13-01708-f001:**
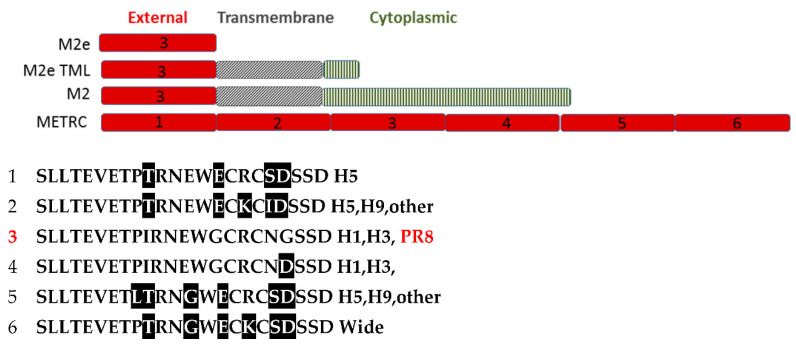
Influenza matrix protein M2 structural features and comparison to METRC used in this study. M2e domains (red in diagram) are the external 23 amino acids of the M2 molecule. An M2e TML construct contained M2e from PR8 H1N1 and a conserved transmembrane region with an additional 16 cytoplasmic amino acids for a total of 58 amino acids. The M2 molecule in nature consists of an external domain (red, M2e), a transmembrane domain (slashes, grey), and a cytoplasmic domain (lined, green) for a total of 97 amino acids. METRC contains 6 versions of M2e domains (red), each found in nature. Amino acid sequences of each version of M2e found in METRC are listed with amino acid substitutions relative to peptide #3 which is native to PR8.

**Figure 2 viruses-13-01708-f002:**
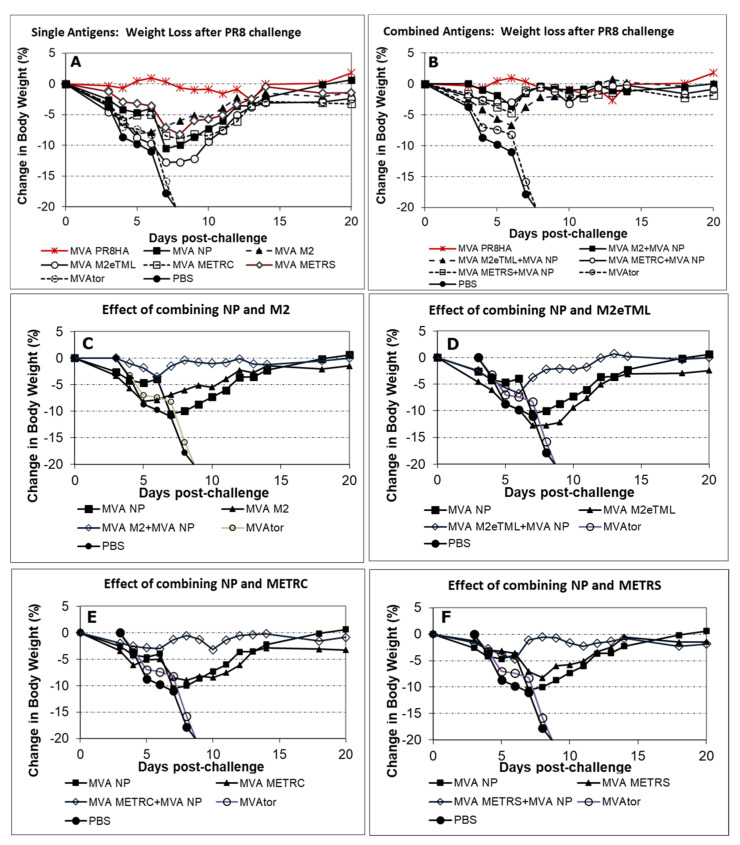
Body weight loss after PR8 H1N1 challenge of mice vaccinated with MVA-based vaccines. Combined conserved non-HA antigen MVA vaccines resulted in weight loss similar to an HA MVA vaccine. Mice were vaccinated intramuscularly two times, days 0 and 18–22, and challenged intranasally on day 42. (**A**) HA was superior to other single non-HA antigens. (**B**) Combined antigen vaccines resulted in less than 7% weight loss. (**C**) Data for NP and M2 vaccines, alone and combined. (**D**) Data for NP and M2eTML vaccines, alone and combined. (**E**) Data for NP and METRC vaccines, alone and combined. (**F**) Data for NP and METRS vaccines, alone and combined.

**Figure 3 viruses-13-01708-f003:**
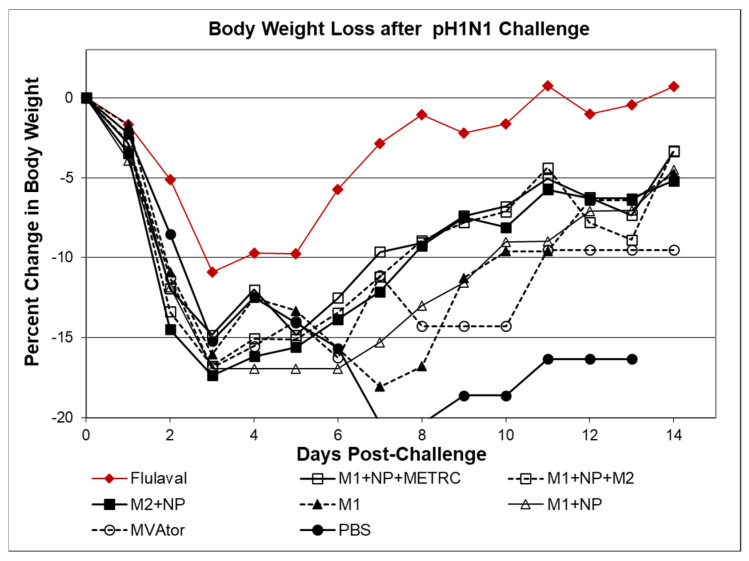
Change in body weight and survival of BALB/c mice challenged with six LD50 (LD99) pH1N1. Eight groups of mice (*N* = 8) either immunized with various MVA-vector based vaccines or with MVA vector (MVAtor) were challenged with pH1N1 3 weeks after the final immunization. Daily body weights were monitored up to 14 days post-challenge. Weight loss >20% body weight was scored as death. Vaccines that delivered NP combined (blended) with M2 or METRC had similar (15%) weight loss that improved over the monitored period. Despite substantial weight loss in recipients of single antigen MVA-M1 vaccine, M1 did not lessen the benefit of blended MVA M1 + MVA NP + MVA M2 or blended MVA M1 + MVA NP + MVA METRC vaccines.

**Figure 4 viruses-13-01708-f004:**
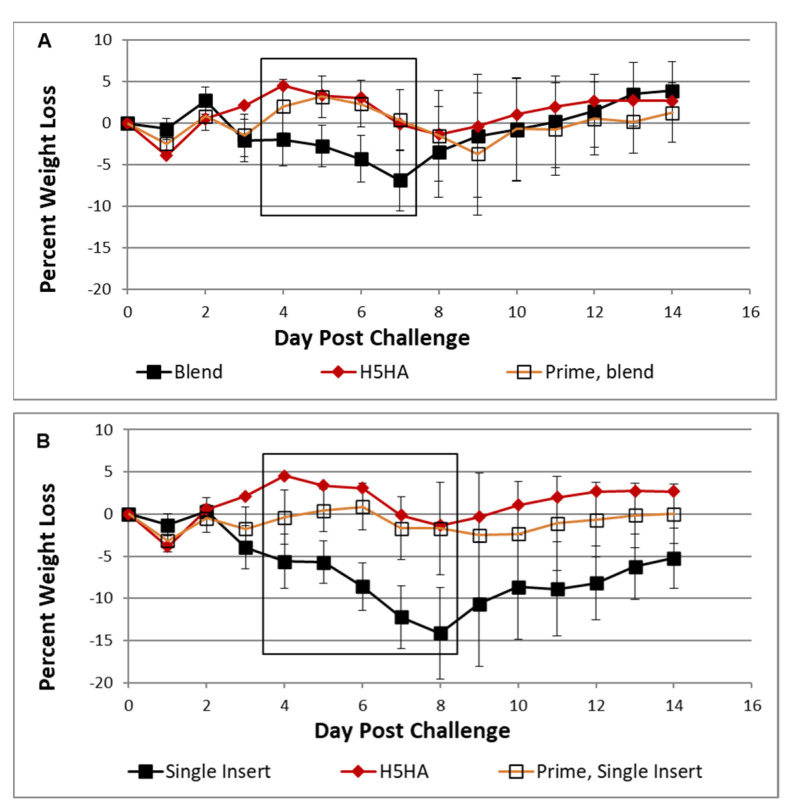
Based upon body weight of BALB/c mice after H5N1 challenge, non-HA antigen vaccines can build upon immunity conferred previously by natural heterologous infection (empty squares), reaching the level of weight loss protection provided by H5 HA (solid red diamonds) delivered two times. Mice in groups of eight were immunized two times using MVA vaccines (solid symbols) or after natural INFV exposure (“Prime”, empty symbols) using low dose non-lethal PR8 INFV A. After H5N1 lethal challenge, mice were monitored for daily body weight and survival for 14 days. The period of 4 to 7–8 days post challenge is framed in each panel to illustrate absence of weight loss due to natural INFV exposure. Heterologous exposure to native PR8 H1N1 INFV A, followed by administration of NP + METRC + M1 expressed as a single insert in MVA ((**A**), empty squares, “Prime, single insert”) or as a blended vaccine of individually expressed antigens in MVA ((**B**), empty squares, “prime, blend”) conferred minor weight loss similar to homologous HA vaccine.

**Figure 5 viruses-13-01708-f005:**
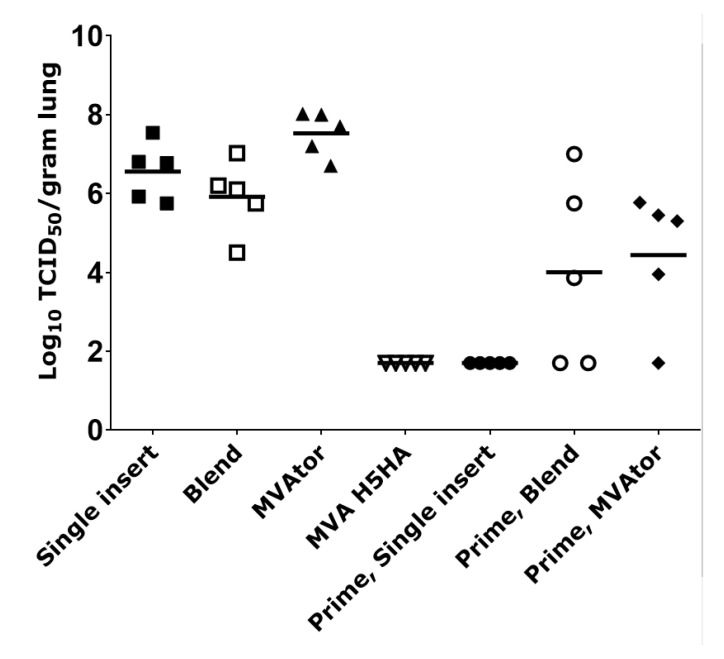
Lung viral burdens after immunization and H5N1 challenge. Mice were immunized two times using a single MVA vaccine that encoded M1 + NP + METRC (“single insert”) or a combination of three separate MVA vaccines each encoding M1 or NP or METRC (“Blend”). “Prime” indicates previous mild infection with PR8 H1N1 INFV A, 21 days prior to vaccine administration. A TCID_50_ assay was performed to quantify infectious H5N1 virus in lungs (*N* = 5 per group) 4 days after A/VN/1203/04 challenge (H5N1). The horizontal bar for each group represents the arithmetic mean Log_10_ TCID_50_ per gram tissue. Individual dots represent the results of each animal. The limit of detection for this assay was 1.7 Log_10_ TCID_50_. Lung viral burden on day 2 (data not shown) had nearly identical results compared to day 4. Without PR8 priming, only the MVA H5HA vaccine was different (*p* < 0.0001, ANOVA followed by Tukey–Kramer test) from the MVAtor control. If delivered after PR8 priming, MVA encoding M1 + NP + METRC as a single insert again was significantly different from MVAtor with priming (*p* = 0.019). Priming gave protective advantage to all test vaccines including the MVAtor negative control. Limit of detection was log 1.7.

**Figure 6 viruses-13-01708-f006:**
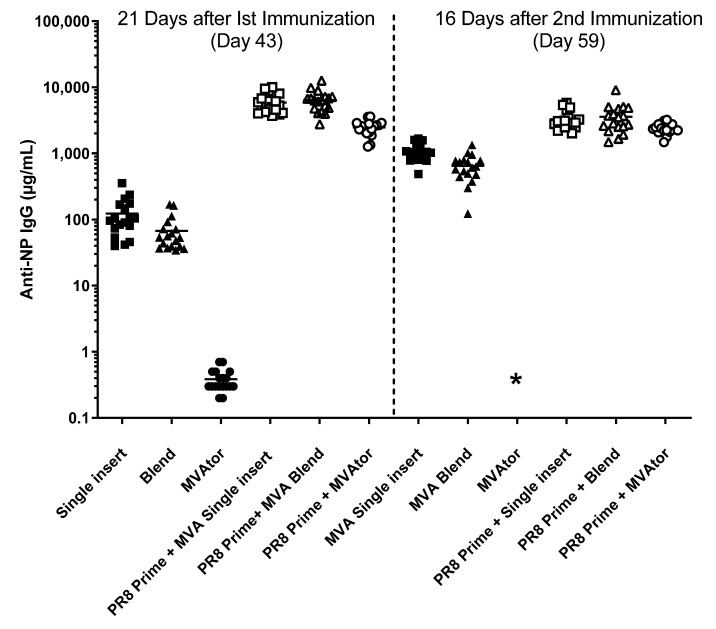
Levels of anti-NP IgG in sera of mice after vaccination with MVA vaccines that encoded NP combined with other antigens. Levels of serum anti-NP IgG were analyzed by ELISA. “Prime” refers to intranasal instillation of sublethal PR8 on day 0; first immunization using an MVA vaccine occurred on day 22; second immunization using MVA vaccines occurred on day 43. MVAtor alone did not result in serum anti-NP IgG antibody levels but priming using PR8 generated high IgG anti-NP levels. Squares depict single insert encoding M1 + METRC + NP; triangles depict a blend of individual M1, METRC and NP constructs; circles depict MVAtor controls. Filled symbols depict MVA vaccine treatments; empty symbols depict non-lethal (10% weight loss) PR8 infection as a priming event. * Anti-NP IgG ELISA were not performed using sera of this group.

**Figure 7 viruses-13-01708-f007:**
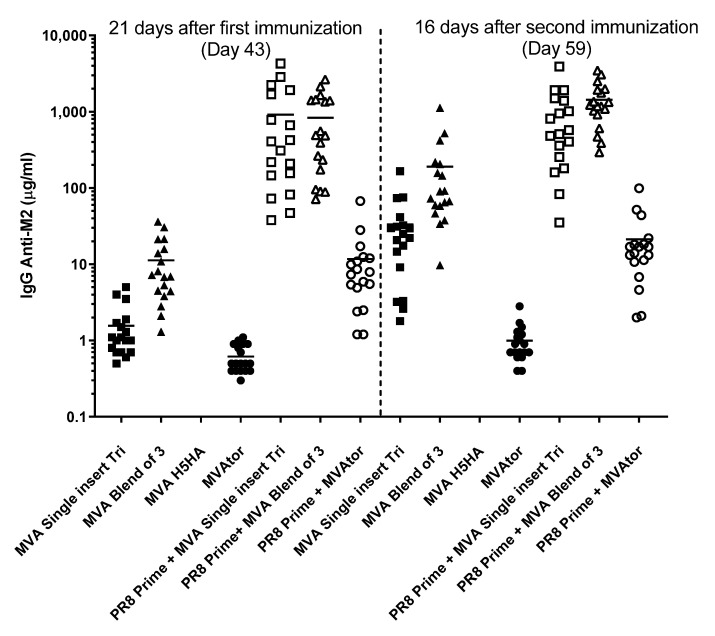
Anti-M2 IgG immune response in mice that received vaccines that encoded METRC antigen (Immunization days: Optional prime day 0, MVA Day 22 and Day 43). “PR8 Prime” refers to intranasal installation of sublethal PR8 on day 0; first immunization using an MVA vaccine occurred on day 22; second immunization using MVA vaccines occurred on day 43. For virus-primed mice, serum anti-M2 IgG antibody levels did not increase beyond the primed and boosted level despite an additional immunization. Squares depict single insert encoding M1 + METRC + NP; triangles depict a blend of individual M1, METRC and NP constructs; circles depict MVAtor controls. Filled symbols depict MVA vaccine treatments; empty symbols depict non-lethal PR8 infection as a priming event.

**Table 1 viruses-13-01708-t001:** Survival of influenza challenge after two immunizations. Numbers of survivors per group of 8 mice are listed as percent (%) survival.

MVA-Encoded Vaccine Antigens	Challenge		
	PR8 H1N1	pH1N1	H5N1
MVAtor	0/8 (0%)	1/8 (13%)	0/8 (0%)
PBS	0/8 (0%)	2/8 (25%)	
NP	8/8 (100%)		
M2	8/8 (100%)		
M2TML	8/8 (100%)		
METRC	8/8 (100%)		
METRS	8/8 (100%)		
NP + M2	8/8 (100%)		
NP + M2TML Bl ^1^	8/8 (100%)		
NP + METRC Bl ^1^	8/8 (100%)		
NP + METRS Bl ^1^	8/8 (100%)		
M1		3/8 (38%)	
NP + M1 Bl ^1^		7/8 (88%)	
NP + M2 Bl ^1^		8/8 (100%)	
NP + M2 + M1 Bl ^1^		8/8 (100%)	
NP + METRC + M1 Bl ^1^		8/8 (100%)	7/8 (88%)
NP + METRC + M1 SI ^2^			4/8 (50%)
H5 (VN) HA			8/8 (100%)
MVAtor			0/8 (0%)
PR8 Prime, NP + METRC + M1 Bl ^1^			8/8 (100%)
PR8 Prime, NP + METRC + M1 SI ^2^			8/8 (100%)
PR8 Prime, MVAtor			8/8 (100%)

^1^ Bl, blend of separate vaccine constructs. ^2^ SI, single insert that contained genes of three antigens.

**Table 2 viruses-13-01708-t002:** HAI antibody responses to MVA PR8 HA vaccine were similar to sublethal infection by PR8 virus, neither of which could be boosted by further exposure to MVA HA vaccine or INFV A challenge. Initial exposure to virus or MVA HA(PR8) vaccine resulted in weak HAI titers (column #1) that were boosted by re-administration (column #2). HAI titers were not boosted further (column #3) by additional exposure to MVA HA (PR8) vaccine or lethal PR8 viral challenge. Antibody responses were measured as hemagglutination inhibition (HAI) titers. The HAI test used PR8 influenza virus. Group mean HAI titers are shown over two experiments.

Immunization or Challenge →	1	2	3
Experiment One: two administrations of sublethal INFV A followed by sublethal INFV A or lethal INFV A	Sublethal Influenza *N* = 25 mice HAI titer = 436 ± 141	Sublethal Influenza *N* = 25 mice HAI titer = 1173 ± 478	Sublethal Influenza *N* = 5 mice HAI titer = 1440 ± 556
			Lethal Challenge *N* = 16 HAI titer = 1307 ± 596
Experiment Two: two administrations of MVA HA vaccine, followed by MVA HA or lethal INFV A	MVA HA *N* = 25 mice HAI titer = 128 ± 535	MVA HA *N* = 25 mice HAI titer = 1182 ± 653	MVA HA *N* = 5 mice HAI titer = 981 ± 324
			Lethal Challenge *N* = 16 HAI titer = 953 ± 458

**Table 3 viruses-13-01708-t003:** Proteins and peptides utilized as recall antigens for ELISpot analysis of splenocytes obtained after delivery of MVA vaccines that encoded NP, M2 or M2e. *p* values were calculated by *t*-test, 2-tailed, equal variance. Influenza recall antigens were derived from proteins of H1N1 PR8 INFV A.

Statistical Difference Compared to Medium	Recall Antigen	Antigen	MHC Class	Spots Per 300,000 Cells
	Medium	None	None	3 ± 2
*p* < 0.001	NP (H1N1) protein	NP	NA ^1^	98 ± 56
*p* < 0.001	TYQRTRALV peptide	NP	I	325 ± 126
*p* < 0.001	RLIQNSLTIERMVLS peptide	NP	II	53 ± 31
None	M2 protein (H1N1 PR8)	M2	NA ^1^	11 ± 17
None	IAANIIGIL peptide ^2^	M2	I	7 ± 11
None	IYRRFKYGL peptide ^2^	M2	I	3 ± 2
None	LLTEVETPI peptide	M2e	I	7 ± 11
*p* < 0.001	VETPIRNEW peptide	M2e	I	38 ± 18
*p* < 0.001	SLLTEVETPIRNEWGCRCNGSSD peptide	M2e	II	26 ± 18
NA ^3^	SPGAAGWDL and VGPSNSPTF	MVA	I	666 ± 71

^1^ Not applicable, recall antigen was whole-molecule protein. ^2^ Peptides IAANIIGIL and IYRRFKYGL are within the internal domains of the M2 molecule, and are not found within the external M2e domain, therefore were not contained within the vaccines tested. These low and high scoring (respectively) epitope peptides [28] are negative controls. ^3^ Not applicable. MVA MHC class I peptides [29] were system suitability standards.

## Data Availability

Not applicable.

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
