# Peer review of "Influenza Antigens NP and M2 Confer Cross Protection to BALB/c Mice against Lethal Challenge with H1N1, Pandemic H1N1 or H5N1 Influenza A Viruses"

_viruses, 2021, doi:10.3390/v13091708_

Round 1
Reviewer 1 Report
The authors excessively assessed protective efficacy of various modified Vaccinia virus Ankara (MVA)-vectored Influenza vaccines in mice. It is scientifically sound and contains sufficient interest and originality to merit publication. However, I would suggest extensive revision in combination with re-review for this manuscript. Major points: 1. Please add statistical method and analysis for Figures and Tables where applicable. 2. Section 2.3: To confirm the expression of proteins, immunofluorescent staining is not enough as you can’t confirm the size of the protein. Also how the authors confirm the M2e constructs? Especially for METRC constructs I think you need antibody specific to each M2e sequence and perform western blotting to confirm the size. 3. Table 2 must be completed or at least show the protective efficacies of single antigens (NP, M2 or PR8) immunization against all three viruses (PR8 H1N1, pH1N1 & H5N1). With current result, it looks like NP or PR8 is enough to protect against all three virus challenge. 4. Also Table 2 shows PR8 priming is enough to protect against H5N1 challenge. But will this true in Flu-infected Humans? 5. Please explain about Flulaval. 6. Figure 4: I don’t think “The time period of 4 to 8 days post challenge is outlined in a box in each panel.”. 7. Line 383: Please include the reference for “Anti-NP IgG response was a sensitive indicator of immunity”. 8. Line 436: Please include anti-NP antibody level for PR8, MVAtor group. 9. Line 446: Please discuss about the mechanism of protection by anti-NP antibody. Minor points: 1. Line 27: “antigenic rift and antigeni shift” should be “antigenic drift and antigenic shift” 2. Line 152: “human euthanasia” should be “humane euthanasia” 3. Figure 2B: Legend for PBS is missing. 4. Figure3: Two graphs are overlapping each other and they are not in left & right orientation. 5. Line 467: “H2-d” should be “H-2d”.Author Response
Thank you for your detailed review and your comments. We have responded to the suggestions and requirements. The responses are itemized in the attached WORD document. In addition, many of the requested changes were made in red font in the new version of the manuscript.

Reviewer 2 Report
In this manuscript, the authors assessed the ability of MVA vaccines to induce protective immunity against conserved epitopes of influenza viruses. The vaccines and manuscript are generally of great interest, but the reader is left to desire a bit more, as mechanism of protection and true cross-reactivity aren't well established. Minor revisions are required, but several experiments are requested to improve the clarity of the paper (listed below). Also, the paper was a bit difficult to read, so additional editing of the text throughout would be greatly appreciated to improve clarity. Specific examples are listed below.
Major comments
- Line 99 – are these antigens retained intracellularly? Secreted? Or confirmed to be expressed on the cell surface?
- The authors used flulaval in Figure 3, presumably to allow them to compare their vaccine to that of an approved vaccine. It is unknown what the formulation of this vaccine ism, the dosing, use of adjuvants, route of administration, etc.
- I like this control, but would have liked to see it for the PR8 and H5 challenge studies. Those data would have supported the authors’ claims as to why it is so important to target internal/conserved antigens.
- The authors speculate that serum antibodies against these conserved antigens are protective. However, their role in protective immunity are quite controversial and further experimentation would be appreciated. Serum transfer prior to infection or immunization of muMT (near B cell knockout) + infection would provide further insight into the role of B versus T cells.
- These antigens are pretty well conserved across influenza subgroups. Did the authors test against group 2 viruses (H3, H7, etc)?
Minor comments
- Line 27 – a few letters are missing on antigenic drift and antigenic shift
- Line 27 – a bit of an over generalization that antigenic shift necessitates annual immunizations, as they occur very rarely.
- Line 27 and throughout – please use the full terminology for influenza or the abbreviation, not flu. Also when discussing results related to influenza viruses, please state influenza viruses (or the abbreviation included in the paper), not influenza. Influenza is the disease, the influenza virus is the pathogen that causes the disease. The authors are analyzing immunity against the virus, not the disease.
- Line 49 – 16 NA subtypes? There are only 11, including the two distantly related bat NAs.
- Line 50 – this sentence is very confusing. I think the authors are trying to suggest that the vaccine has limitations due to antigenic drift and shift. Breaking this sentence into two and framing these points around these other studies would be helpful.
- Line 61 – reference(s) for antibody responses against more conserved antigens mediating broader immunity and cross-protection?
- Line 65 – required for protection against influenza? Stated weird as is.
- Figure 4 appears in the text before Figure 3.
- Figure 3 figure legend – The authors don’t break the panels up into A or B, but rather left or right. The panels are stacked on top of one another, so left and right isn’t applicable.
- Please define what each symbol in Figure 6 and 7 indicate.
- Have the peptides used for the ELISPOT been published before? If so, what is the reference? If not, how did the author settle on these peptides?
- T cell ELISpot results should have a graph near the data, not in the methods.
- Line 464 and 465 – CTL is not defined, which is cytotoxic T lymphocyte aka CD8 T cells. Those are T cell immunity, so this sentence is confusing.
Author Response
Thank you for your detailed comments. We have proceeded through each item and listed them in the WORD docoument attached. In addition, changes have been made in the manuscript, many in red font to allow ease of recognition.

Round 2
Reviewer 1 Report
The manuscript is well revised.
please amend Line 504: PR9 to PR8 and ""mcg" to "µg" (Lines 400, 405, 512)
Author Response
Reviewer 1 made two request in round 2:
- please amend Line 504: PR9 to PR8
- and "mcg" to "µg" (Lines 400, 405, 512)
These items have been corrected. In addition the author has changed "mcg" to "µg" in lines 221, 404, 536
Also, the author changed red font text to blank font. Red had been used to indicate changes and additions of text in round 1 of response to Reviewers.
